

# Asymmetrical dispersal and putative isolation-by-distance of an intertidal blenniid across the Atlantic–Mediterranean divide

Rita Castilho[1], Regina L. Cunha[1], Cláudia Faria[2], Eva M. Velasco[3] and Joana I. Robalo[4]

[1] Center of Marine Sciences (CCMAR), University of Algarve, Faro, Portugal
[2] Instituto de Educação da Universidade de Lisboa, Lisboa, Portugal
[3] Centro Oceanográfico de Gijón, Instituto Español de Oceanografía, Gijón, Spain
[4] ISPA Instituto Universitário de Ciências Psicológicas, Sociais e da Vida, MARE—Marine and Environmental Sciences Centre, Lisboa, Portugal

## ABSTRACT

Transition zones are of high evolutionary interest because unique patterns of spatial variation are often retained. Here, we investigated the phylogeography of the peacock blenny, *Salaria pavo*, a small marine intertidal fish that inhabits rocky habitats of the Mediterranean and the adjacent Atlantic Ocean. We screened 170 individuals using mitochondrial and nuclear sequence data from eight locations. Four models of genetic structure were tested: panmixia, isolation-by-distance, secondary contact and phylogeographic break. Results indicated clear asymmetric migration from the Mediterranean to the Atlantic but only marginally supported the isolation-by-distance model. Additionally, the species displays an imprint of demographic expansion compatible with the last glacial maximum. Although the existence of a refugium in the Mediterranean cannot be discarded, the ancestral lineage most likely originated in the Atlantic, where most of the genetic diversity occurs.

## INTRODUCTION

Many terrestrial and marine species have often experienced expanding and contracting range shifts over time (*Herborg et al., 2007*; *Reece et al., 2010*; *Reuschel, Cuesta & Schubart, 2010*). These range shifts are generally promoted by geological or climate events that affect temperature and territorial connectivity between locations. The African and Iberian continental margins formed the Gibraltar arch at 5.5 MYA (million years ago) producing a land bridge that interrupted the water flow between the Atlantic and Mediterranean adjacent basins. This event, known as the Messinian Salinity Crisis, turned the Mediterranean hypersaline and dried out large expanses of the basin (*Duggen et al., 2003*; *Hsü, Ryan & Cita, 1973*; *Krijgsman, 2002*). The Mediterranean Sea was then the ground of a drastic contraction-expansion of distributional range processes in marine organisms inhabiting

Corresponding author
Rita Castilho, rita.castil@gmail.com

those waters. The disappearance of the previously existing Tethyan fauna followed by the Mediterranean invasion of Atlantic species through the Strait of Gibraltar, when the land bridge receded, were the main drivers of those distributional processes. Furthermore, the Pleistocene glacial episodes and the consequent fluctuations of the sea level and surface temperature in the Mediterranean and adjacent Atlantic have shaped the distribution of marine organisms impacting their genetic makeup (*Patarnello, Volckaert & Castilho, 2007*).

The Mediterranean Sea and the contiguous Northeastern Atlantic Ocean were the focus of several phylogeographic studies on marine fish exploring the relationships between populations inhabiting both regions across a well-defined oceanographic break, the Almeria-Oran Front (AOF). The AOF is situated east of the Strait of Gibraltar by the convergent Atlantic and Mediterranean water masses, stretching from Almeria on the Spanish coast to Oran on the Algerian coast. Some species such as *Dicentrarchus labrax*, *Diplodus puntazzo* and *Coryphoblennius galerita* have shown high genetic divergence between populations inhabiting both sides of the AOF (*Bargelloni et al., 2005*; *Domingues et al., 2007*; *Lemaire, Versini & Bonhomme, 2005*) while others display evidence of strong genetic flow (e.g., *Thalassoma pavo*, *Chromis chromis* and *Diplodus sargus*—*Bargelloni et al., 2005*; *Costagliola et al., 2004*; *Domingues et al., 2005*). It has proven difficult to assign these differences to a single environmental or biological parameter (e.g., *Galarza et al., 2009b*).

This study is part of an on-going effort to understand the phylogeography of intertidal fish fauna with their areas of distribution centred in the Northeastern Atlantic Ocean and the adjacent Mediterranean Sea (e.g., *Almada et al., 2012*; *Francisco et al., 2009*; *Robalo et al., 2012*). Species with such distribution, different habitat requirements, and diverse larval pelagic durations, provide interesting opportunities to study the evolutionary effects of geographic range shifts and genetic patterns of differentiation. Present in both areas, the peacock blenny, *Salaria pavo*, occurs mainly around the Western Mediterranean coasts and from the Bay of Biscay south to the Canaries (*Zander, 1986*), being less abundant in the Eastern Mediterranean. This species lives in sheltered rocky habitats and coastal lagoons, in the intertidal, or in the first meters of the subtidal. Contrary to other Blenniidae, the peacock blenny is able to colonize soft substrates (mud and sandy bottoms) and isolated patches of underwater vegetation (*Verdiell-Cubedo, Oliva-Paterna & Torralva, 2006*). *Salaria pavo* displays a high tolerance to salinity (from 2 to 65‰) and temperature from 1° to 30 °C (*Paris & Quignard, 1971*; *Plaut, 1999*). Nevertheless, the ecology of the species varies enormously with the availability of spawning grounds (e.g., *Almada et al., 1994*, and references therein). During the spawning season, males build and defend nests from conspecific males or other intruders and care for the eggs (*Gonçalves & Almada, 1997*). Eggs of *S. pavo* are unable to hatch at temperatures below 15 °C and typically breed at temperatures above 18 °C (*Westernhagen, 1983*). Therefore, at the Last Glacial Maximum (LGM), when temperatures are estimated to have been between 1.5 °C in February and 9.5 °C in August, suitable temperatures for the reproduction of *S. pavo* were likely absent from the Bay of Biscay, western Galicia and northern Portugal.

The goal of this study was to investigate genetic imprints of the peacock blenny, *Salaria pavo*, using the mitochondrial D-loop and the first intron of the nuclear S7 ribosomal protein gene sequences. More specifically, we assessed the genetic diversity and population

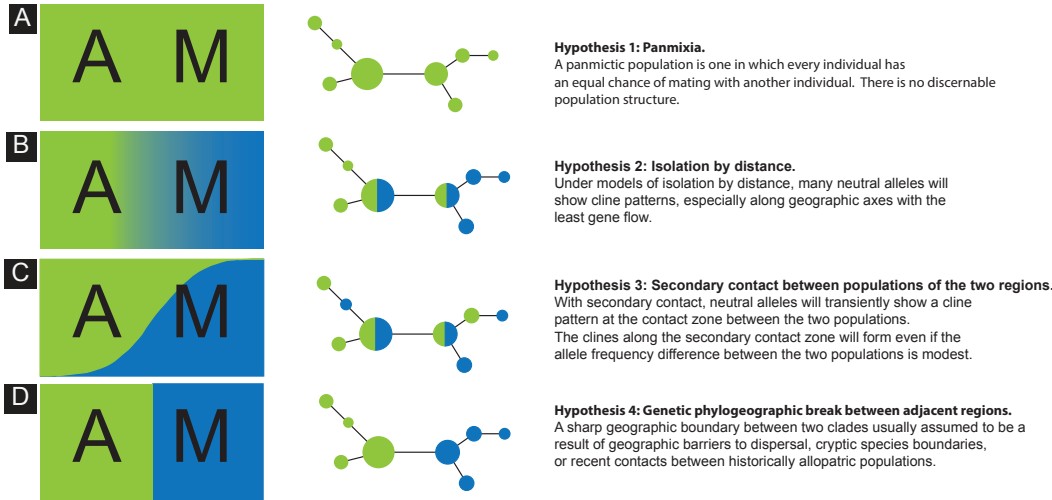

**Hypothesis 1: Panmixia.**
A panmictic population is one in which every individual has an equal chance of mating with another individual. There is no discernable population structure.

**Hypothesis 2: Isolation by distance.**
Under models of isolation by distance, many neutral alleles will show cline patterns, especially along geographic axes with the least gene flow.

**Hypothesis 3: Secondary contact between populations of the two regions.**
With secondary contact, neutral alleles will transiently show a cline pattern at the contact zone between the two populations. The clines along the secondary contact zone will form even if the allele frequency difference between the two populations is modest.

**Hypothesis 4: Genetic phylogeographic break between adjacent regions.**
A sharp geographic boundary between two clades usually assumed to be a result of geographic barriers to dispersal, cryptic species boundaries, or recent contacts between historically allopatric populations.

**Figure 1  Schematics of four models.** Schematics of four models of haplotype frequency distribution and haplotype networks that are expected to result from the scenarios involving panmixia (A), isolation-by-distance (B), secondary contact (C) and phylogeographic barrier (D).

structure of this species over its sampled distribution range and evaluated long-term connectivity among populations. We evaluated the following biogeographic hypotheses concerning the current spatial genetic diversity of *S. pavo*: (1) panmixia, whereby there is no discernible geographic or otherwise genetic structure corresponding effectively to a random distribution of haplotypes (Hypothesis 1: Fig. 1A); (2) isolation-by-distance (IBD) pattern by which genetic and geographic distances are positively correlated (*Wright, 1943*), and therefore alleles will show a frequency cline pattern between the Atlantic and the Mediterranean (Hypothesis 2: Fig. 1B); (3) secondary contact between populations of the two regions, where alleles will transiently show a cline pattern at the contact zone between the two areas (Hypothesis 3: Fig. 1C), and (4) genetic phylogeographic break between adjacent regions, wherein a sharp change of allele frequencies is observed between the Atlantic and the Mediterranean (Hypothesis 4: Fig. 1D).

## MATERIAL AND METHODS

### Sampling and generation of molecular data

Samples of *S. pavo* were collected at 8 localities in the Northwestern Mediterranean and Atlantic coast of the Iberian Peninsula (Table 1; Fig. 2). No field permits were required as this species is listed as "least concern conservation status" and it was not captured in protected areas. Fish were caught by scuba diving and small fishnets on rocky beaches and fin clips were stored individually in 96% ethanol. Captured fish were held in buckets with aerators, manipulation kept to the minimum time required to collect the smallest fin clip sample possible and quickly placed back in the live-cart prior to release. Fish were observed for general condition before release, and returned to the same pond or location of capture as soon as possible.

Castilho et al. (2017), *PeerJ*, DOI 10.7717/peerj.3195

**Table 1** Sample locations, sample abbreviation code, sample sizes and summary statistics for a sequence fragment of the mtDNA D-loop and the first intron of S7 nuclear gene of *Salaria pavo*.

| Region | Locations | Code | Mitochondrial D-loop | | | | | | | First intron of S7 gene | | | | |
| | | | $N$ | NH | NP | Haplotype diversity ± s.d. | Nucleotide diversity ± s.d | PS | $N$ | NH | Gene diversity ± s.d. | Nucleotide diversity ± s.d | Observed Heterozygosity |
|---|---|---|---|---|---|---|---|---|---|---|---|---|---|
| M | Barcelona | BA | 16 | 9 | 3 | 0.92 ± 0.04 | 0.0520 ± 0.0274 | 38 | 15 | 5 | 0.65 ± 0.07 | 0.0017 ± 0. 0013 | 0.53 |
| | Formentera | FO | 2 | 2 | 1 | | | | 3 | 1 | | | |
| | Cabo de Gata | CG | 12 | 6 | 2 | 0.88 ± 0.06 | 0.0167 ± 0.0098 | 10 | 13 | 4 | 0.64 ± 0.07 | 0.0015 ± 0.0013 | 0.46 |
| A | Cadiz | CA | 39 | 19 | 13 | 0.88 ± 0.04 | 0.0151 ± 0.0084 | 25 | 22 | 6 | 0.72 ± 0.05 | 0.0019 ± 0.0015 | 0.41 |
| | Ria Formosa | RF | 23 | 13 | 8 | 0.94 ± 0.03 | 0.0111 ± 0.0066 | 17 | 40 | 6 | 0.62 ± 0.04 | 0.0016 ± 0.0013 | 0.45 |
| | Olhos de Água | OA | 12 | 7 | 6 | 0.91 ± 0.08 | 0.0170 ± 0.0100 | 18 | 12 | 5 | 0.78 ± 0.07 | 0.0023 ± 0.0017 | 0.42 |
| | Sado | SA | 26 | 7 | 4 | 0.72 ± 0.08 | 0.0145 ± 0.0083 | 12 | 30 | 4 | 0.73 ± 0.03 | 0.0019 ± 0.0014 | 0.40 |
| | Galicia | GA | 1 | 1 | 1 | | | | 1 | 1 | | | |

**Notes.**

N, number of individuals per location; NH, haplotype richness; NP, number of private haplotypes; PS, number of polymorphic sites.
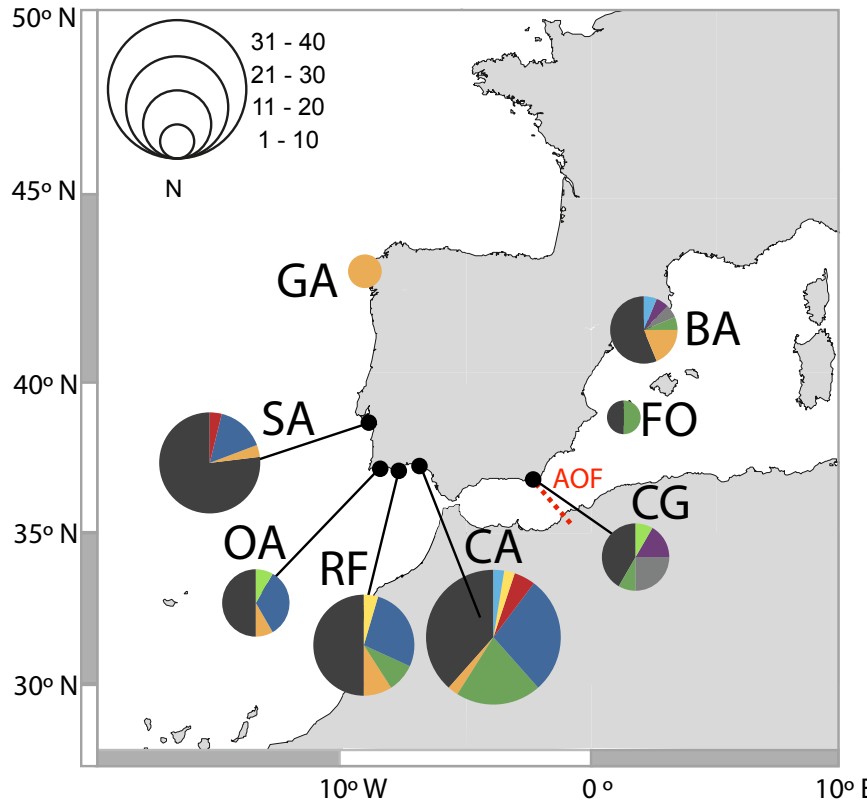

**Figure 2** **Distribution of D-loop haplotypes of *Salaria pavo* on each location.** Two-letter codes refer to the name of locations in Table 1. Colours allow for comparison of the presence of common haplotypes that are present in locations. The biogeographical break of the Almeria-Oran front is represented in red (AOF).

Total genomic DNA was extracted from fin or muscle samples with the REDExtract-N-Amp kit (Sigma-Aldrich) following the manufacturer's instructions. Voucher specimens are deposited in ISPA (ethanol preserved tissues). We selected two unlinked genes of different genomes to be sequenced: the mitochondrial D-loop and the nuclear S7 ribosomal protein gene (S7, 520 bp, including the first intron (*Chow & Hazama, 1998*)). Nuclear and mitochondrial sequences were obtained from the same individuals whenever possible. PCR amplification of mitochondrial D-loop and of the S7 were performed with the following pairs of primers: D-loop—LPro1 and HDL1 (*Ostellari et al., 1996*) and S7—S7RPEX1F and S7RPEX2R (*Chow & Hazama, 1998*). PCR amplification reactions were performed in a 20 μl total-reaction volume with 10 μl of REDExtract-N-ampl PCR reaction mix (Sigma–Aldrich), 0.8 μl of each primer (10 μM), 4.4 μl of Sigma-water and 4 μl of template DNA. An initial denaturation at 94 °C for 7 min was followed by 35/30 cycles (denaturation at 94 °C for 30/45 s, annealing at 55 °C for 30/45 s, and extension at 72 °C for 1 min) and a final extension at 72 °C for 7 min on a BioRad MyCycler thermal cycler (values D-loop/S7, respectively). The same primers were used for the sequencing reaction, and the PCR products were purified and sequenced in STABVIDA (http://www.stabvida.net/). Sequences for each locus were aligned, edited, and trimmed to a common length using the

DNA sequence assembly and analysis software GENEIOUS PRO 7.0 (Biomatters, LTD, Auckland, NZ).

## Genetic diversity and population differentiation

The gametic phase of multi-locus genotypes of the nuclear S7 intron was determined using the pseudo-Bayesian approach of Excoffier–Laval–Balding (ELB) algorithm (*Excoffier, Laval & Balding, 2003*), as implemented in ARLEQUIN 3.5 (*Excoffier & Lischer, 2010*). Gene diversity for both D-loop and S7 fragments, described as haplotype (h) and nucleotide (π) diversities (*Nei, 1987*), were calculated using ARLEQUIN 3.5 (*Excoffier & Lischer, 2010*). Observed heterozygosity was also estimated for the S7 fragment. In order to compare haplotype diversity values, the statistics and asymptotic confidence intervals derived by *Salicru et al. (1993)* were used for both overall diversity comparison and pairwise comparisons between locations. A median-joining network (*Bandelt, Forster & Röhl, 1999*) was constructed in NETWORK v4.5 (fluxus-engineering.com) to determine the genealogical relationships among haplotypes and to consider their geographical distributions. POWSIM 4.1 (*Ryman & Palm, 2006*) was used to assess the power of the data and the suitability of sample sizes to detect significant pairwise fixation at different $F_{ST}$ values. Simulations were carried out for an effective population size of $Ne = 2,000$ to yield $F_{ST}$ values of 0.01, 0.02, 0.03, 0.04, and 0.05. Although the species may have a larger effective population size, this is not relevant to the analysis because $Ne$ only determines the time necessary to reach the target $F_{ST}$. In all cases, 1,000 replicates were run and the power of the analysis was indicated by the proportion of tests that were significant at $P < 0.05$ based on chi-squared tests using the respective allele frequencies at the locus studied.

Among the different metrics for population genetic differentiation, we chose to report both fixation indexes (such as $F_{ST}$ and $\Phi_{ST}$) and one genetic differentiation index (Jost D) because they represent different properties of population partitioning (*Bird, Karl & Toonen, 2011*). Genetic fixation $F_{ST}$ and $\phi_{ST}$ (*Weir & Cockerham, 1984*) were estimated with $10^3$ replicates in ARLEQUIN 3.5 (*Excoffier & Lischer, 2010*). Significance level was corrected with Bonferroni and Jost's D differentiation (*Jost, 2008*) (a statistic independent of gene diversity) statistics was estimated with diveRsity package 1.9.5 (*Keenan et al., 2013*) and significance of differentiation was assessed through the calculation of 95% confidence limits using a bias corrected method with $10^4$ bootstraps. Formentera and Galicia, with two and one individuals, were not included in the diversity and population structure analysis.

Mobile species subjected to genetic statistical differentiation tests often fail to display minor amounts of population subdivision even if they exist (*Palumbi & Warner, 2003*). Therefore, we used Spatial Analysis of Shared Alleles (SAShA) (*Kelly et al., 2010*) implemented in the MATLAB environment (Mathworks, Inc.) to test hypothesis 1, i.e., determine the extent to which haplotypes are distributed randomly through space. Non-random distributions of haplotypes can be considered departures from panmixia, and occurrence of the same haplotype in different locations can be considered evidence of recent or ongoing gene flow. SAShA generates the observed distribution of geographic distances of each haplotype, as well as a null distribution generated from the same data. SAShA tests for a significant deviation between the arithmetic mean of the observed

distance distribution (ODD) and that of the expected distance distribution (EDD). An ODD significantly less than EDD indicates that alleles are under-distributed, and therefore gene flow is restricted. We tested for significance of the difference between ODD and EDD using $10^4$ permutations.

To test hypothesis 2, whether the geographical pattern of genetic differentiation is caused by isolation by distance (IBD) we ran Mantel tests (*Mantel, 1967*) for pairwise matrices between geographical distances (kilometres) of the shortest marine path among locations and genetic differentiation (measured as $F_{ST}/(1 - F_{ST})$, $\phi_{ST}/(1 - \phi_{ST})$ and $(D/(1 - D))$. Mantel tests (1,000 randomizations) were performed using mantel.xla 1.2.4 (*Briers, 2003*).

To test hypothesis 4, the existence of a phylogeographic barrier dividing the Atlantic from the Mediterranean, an analysis of molecular variance (AMOVA) was used to examine the amount of genetic variability partitioned among that barrier (*Excoffier, Smouse & Quattro, 1992*). AMOVA computes the proportion of variation among groups ($F_{CT}$ and $\phi_{CT}$), the proportion of variation among populations within groups ($F_{SC}$ and $\phi_{SC}$) and the proportion of variation within populations ($F_{ST}$ and $\phi_{ST}$) (except in the case of S7, where the computation of AMOVA is not possible for $F_{ST}$, with unknown gametic phase data). A modification of this hypothesis, allowing to consider the existence of other partitions, was also explored with the spatial structure of genetic variation using a Spatial Analysis of Molecular Variance (SAMOVA) (*Dupanloup, Schneider & Excoffier, 2002*). SAMOVA defines groups of samples that are maximally differentiated from each other. One hundred simulated annealing processes were used for each value of $K$ (number of groups). The SAMOVA was run from $K = 2$ to the value of $K$ that maximizes the value of the $F_{CT}$ statistic.

## Estimation of gene flow

*Salaria pavo* adults are not known to undertake active migrations, therefore, instead of referring to migration rates ($M$), we will refer instead to gene flow ($G$). $G$ and population size parameter ($\theta$) were inferred using the maximum likelihood (ML) in MIGRATE-N ver. 4.2.6 (*Beerli & Felsenstein, 1999*) among Atlantic and Mediterranean locations in order to determine the degree and direction of migrants across the Atlantic-Mediterranean region. Analyses were first run with a full migration matrix in which gene flow was unrestricted between Atlantic and Mediterranean (asymmetric migration, 4 parameters). To explicitly test other models (including panmixia; immigration into Mediterranean; immigration into Atlantic) we built custom matrices representing gene flow conditions. All $G$ and $\theta$ were calculated using $F_{ST}$ estimates and UPGMA as starting points, and taking into account the model of evolution. A Markov Chain Monte Carlo was run for three short chains of $10^4$ trees and two long chains of $10^5$ trees with a burn-in of $10^3$ trees and a static heating scheme with start temperatures of 1.00, 1.50, 3.00 and 6.00. Finally, likelihood scores for all migration models were obtained by thermodynamic integration with Bezier approximation (*Gelman & Meng, 1998*), as implemented in the software. Direct comparison of models was assessed by manually transforming these likelihood scores into Bayes Factors (*Kass & Raftery, 1995*), which was performed using the method described in *Beerli & Palczewski (2010)*. MIGRATE-N was run on Centre for Marine Sciences (CCMAR) Computational

Cluster Facility (http://gyra.ualg.pt/) and on the R2C2 research group cluster facility, provided by the IT department of the University of Algarve.

## Population demography

Past population demography of *S. pavo* was inferred with the Atlantic D-loop data using the coalescent Bayesian skyline plot (BSP) model as implemented in BEAST v. 2.3.1 (*Ho et al., 2005*) employing the Bayesian MCMC coalescent method, a strict clock and the HKY + I + G model of substitution obtained in Modeltest v. 3.7 (*Posada & Crandall, 1998*), using the Akaike information criterion (AIC) (*Akaike, 1974*). Results were visualized in TRACER v. 1.5 (*Rambaut & Drummond, 2007*). The Bayesian distribution was generated using results from two independent run of 100 million MCMC steps obtaining effective samples sizes (ESS) of parameter estimates of over 200. We used a mutation rate of 3.6% per million years calculated in previous studies where geological events were available to calibrate the rate of D-loop divergence in marine fish (*Donaldson & Wilson, 1999*) and in the absence of a clock calibration for the D-loop of *S. pavo* we address the rate uncertainty by assuming two additional higher within-lineage mutation rates of 5% and 10% per million years.

## RESULTS

### D-loop

A total of 131 D-loop sequences (GenBank accession numbers: HQ857214–HQ857383) were obtained. The D-loop data set after alignment consisted of a total of 300 bp comprising 52 polymorphic sites (17%) and 10 (3%) parsimony informative sites. Overall, mtDNA diversity was high, with 49 haplotypes recovered. A large proportion of haplotypes (57%) were singletons, i.e., represented by a single individual. Forty haplotypes (82%) were private, i.e., occurred in only one location. In total, 92% of haplotypes had a frequency lower than five individuals. The most frequent haplotype in the Atlantic was shared by 25 individuals, followed by two haplotypes shared by 13 individuals, one present only in the Mediterranean and the other in both Mediterranean and Atlantic (Fig. 2). Regarding the 11 haplotypes shared among locations (Fig. 2), six include individuals from both Atlantic and Mediterranean sampling sites. The presence of many low-frequency, closely related haplotypes returns high haplotype diversity (0.952 ± 0.0078) and an average nucleotide diversity (3.63% ± 1.84%) of the overall sample, as well as in each locality (Table 1). Haplotype diversity values were not significantly different between locations, according to the test developed by *Salicru et al. (1993)* ($\chi^2 = 7.15$, $p > 0.05$), except Sado which displayed a significantly lower haplotype diversity when compared to Cadiz, Ria Formosa, Olhos d'Água and Barcelona.

The *S. pavo* haplotype network (Fig. 3) has an overall complex pattern of star-like elements. No evident geographic structure could be depicted from this network, i.e., no discernable association between certain haplotypes and locations can be observed. Although the most frequent haplotype was only present in Portugal and Cadiz, the remaining haplotypes from these localities group together with haplotypes from the Mediterranean. The difference between the overall observed distance distribution (ODD) and the expected distance distribution (EDD) of shared alleles rejected the assumption

**Table 2  Comparison of four biogeographic models for *Salaria pavo*.**

| Marker | Models | | ML_Bézier | Log Bayes factor | Probability |
|--------|--------|--|-----------|------------------|-------------|
| mtDNA | Model 1 | Panmixia | −1266.8 | −19.1 | 0.000 |
| | Model 2 | ATL ↔ MED | −1247.7 | −0.0 | 0.940 |
| | Model 3 | ATL → MED | −1250.5 | −2.8 | 0.060 |
| | Model 4 | MED → ATL | −1262.0 | −14.3 | 0.000 |

of panmixia (hypothesis 1) for the D-loop dataset (ODD = 237 km, EDD = 516 km, $p < 0.00001$) (Fig. 4). POWSIM indicated that a $F_{ST}$ of $\geq 0.0248$ (time in generations = 150) could be detected with $\geq 95\%$ confidence (95.5% Fisher's exact test, 96.2% chi-square). When $F_{ST}$ was set to zero (simulating no divergence among samples), the proportion of $\alpha$ error of type I (rejecting null hypothesis when true) was lower than 5%.

Overall mean pairwise genetic differentiation between the main geographical groups (intra-Mediterranean, between the Mediterranean and Atlantic and intra-Atlantic) showed a tendency for higher Atlantic–Mediterranean values (Fig. 5A). Pairwise location genetic differentiation revealed no association between the levels of differentiation and the three geographical groups considered (Fig. 5B). There is no clear indication of a genetic break (hypothesis 4) between the Mediterranean and Atlantic Ocean as pairwise differentiation values were all within the same range. Isolation-by-distance model (hypothesis 2) support was equivocal, the null hypothesis of no correlation between geographic and genetic distances was not rejected using $F_{ST}$ ($r = -0.075$; significance of $z = 0.3960$) and $\phi_{ST}$ ($r = 0.343$; significance of $z = 0.1140$) but was rejected using $D$ ($r = 0.623$; significance of $z = 0.001$). No haplotype frequency cline (hypothesis 3) could be detected as there were only three haplotypes shared between more than 2 locations.

There was no support for hypothesis 4 (in Appendix S1) indicating non-significant results when we use a strict Mediterranean–Atlantic partition ($F_{CT} = -0.009$, $p$-value = 0.733; and $\phi_{CT} = 0.071$, $p$-value = 0.072), or when we use Barcelona as the sole location representative of the Mediterranean Sea ($F_{CT} = -0.027$, $p$-value = 0.822; and $\phi_{CT} = 0.209$, $p$-value = 0.159). The SAMOVA results showed that $F_{CT}$ statistic does not increase as the number of groups increased (in Appendix S1). The location arrangements that maximized $F_{CT}$ were: 2-group (BA)(CG-CA-RF-OA-SA); 3-group (BA)(CG-CA-RF-OA)(SA); 4-group (BA)(CG)(CA-RF-OA)(SA) and finally 5-group (BA)(CG)(CA)(RF-OA)(SA). None of the $F_{CT}$ values were significant, however, the 4-groups arrangement was marginally non-significant ($p$-value = 0.0583).

MIGRATE-N was run to determine the level and direction of gene flow across the Almeria-Oran oceanographic boundary. The estimated log Bayes factors based on the Bezier approximation score indicated that the most probable model is the one that contemplates asymmetric migration between the Atlantic and the Mediterranean (Table 2). The number of migrants from the Mediterranean to the Atlantic was ca. three times the number of migrants in the inverse direction.

The Bayesian skyline plot indicated that the Western Mediterranean and Atlantic locations of *S. pavo* have experienced a long period of demographic stability in the past,
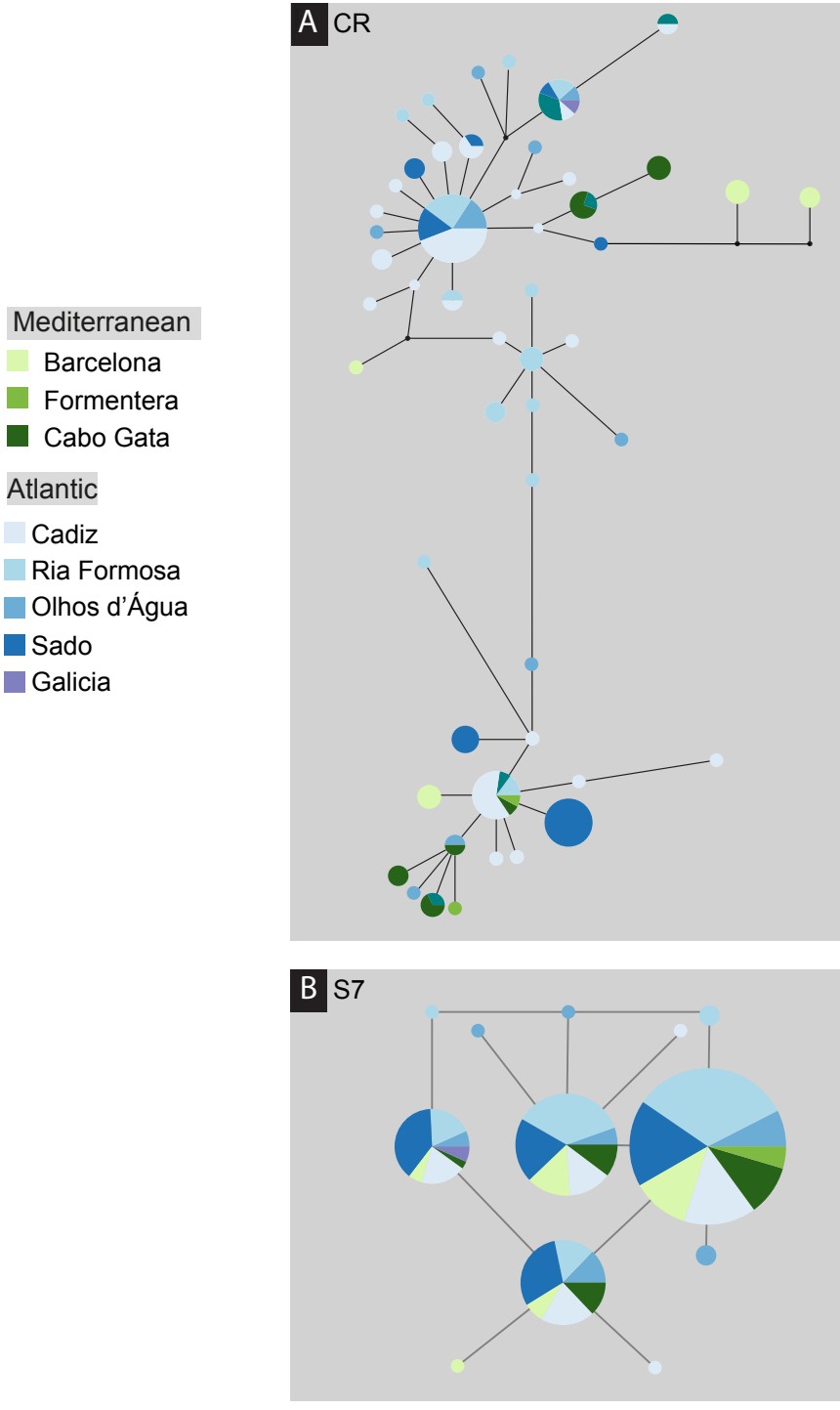

**Figure 3  Median-joining post-processed haplotype network for *Salaria pavo*.** Median-joining post-processed control region (A) and S7 (B) haplotype networks for *Salaria pavo*. The area of the circles is proportional to the frequency of individuals in the sample. Lines are proportional to mutations. Black dots represent median-vectors, or putative haplotypes not sampled or extinct. Colours represent collection location (see key).

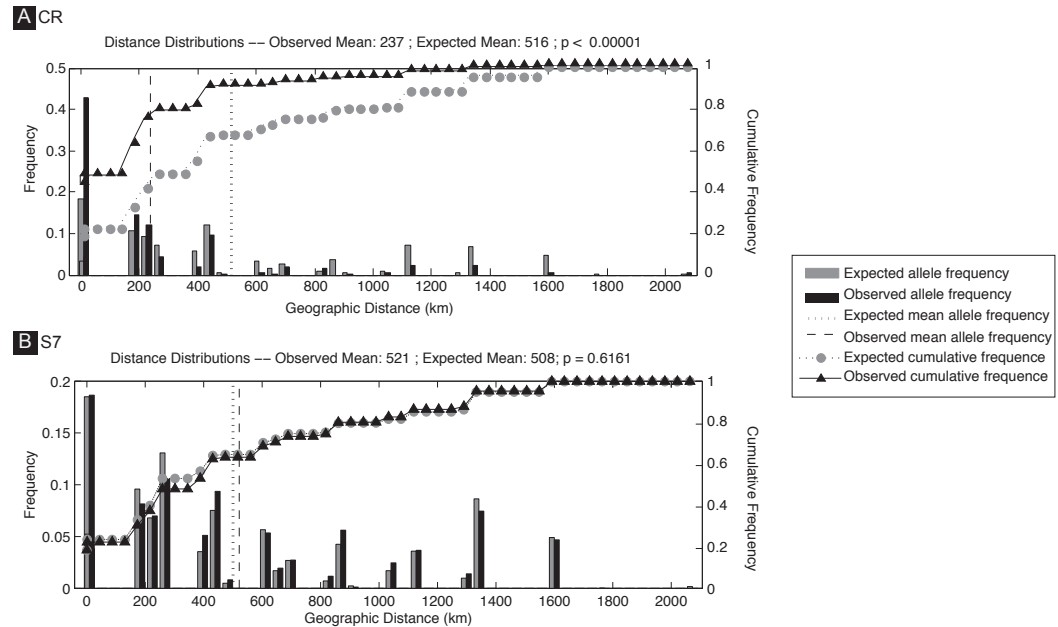

**Figure 4** **Spatial analysis of shared CR mtDNA distribution of *Salaria pavo*.** Spatial analysis of shared alleles for control region (A) and S7 (B). The geographic distances observed between co-occurring alleles and those expected under panmixia are given in the form of histograms and as cumulative frequency plots. The observed and expected mean distances are indicated with vertical lines. Vertical lines represent the observed and expected mean distances. Triangles and circles are the cumulative frequency of alleles at increasing distance. *p*-value is the probability that the observed mean is greater than the expected.

followed by a mild decrease of population size and a quick expansion (Fig. 6). The plot indicates a pronounced ca. 100-fold demographic expansion event. The timeframe of this expansion event is totally dependent on the mutation rates used. Rates of 3.6, 5 and 10%/MY result in expansion dates of 56,000 years, 40,000 years and 20,000 years ago respectively.

## First S7 intron

A total of 136 S7 first intron sequences (GenBank accession numbers: JF834709–JF834885) were obtained. The S7 nuclear region data set after alignment consisted of 519 characters, with seven polymorphic sites, among which five had ambiguities. Using the ELB algorithm, we defined 12 closely related alleles, with four abundant and almost ubiquitous alleles, and the remainder represented by only one or two individuals. Overall gene and nucleotide diversities were low, $0.69 \pm 0.02$ and $0.18\% \pm 0.14\%$, respectively. The haplotype network (Fig. 2) evidences a lack of geographical structure detected by S7, corroborated by both the AMOVA analysis, which returned the proportion of variation among Mediterranean and Atlantic non-significant ($\phi_{CT} = -0007$, *p*-value $= 0.618$) (Appendix S1). Also pairwise comparisons between locations, which returned three significant values, always involving the Ria Formosa location (Ria Formosa—Cabo de Gata; Ria Formosa—Cadiz; Ria Formosa—Sado) (Appendix S1) and non-significant values for the isolation-by-distance Mantel test (Appendix S1). The difference between the overall observed distance
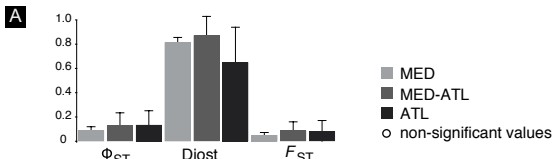

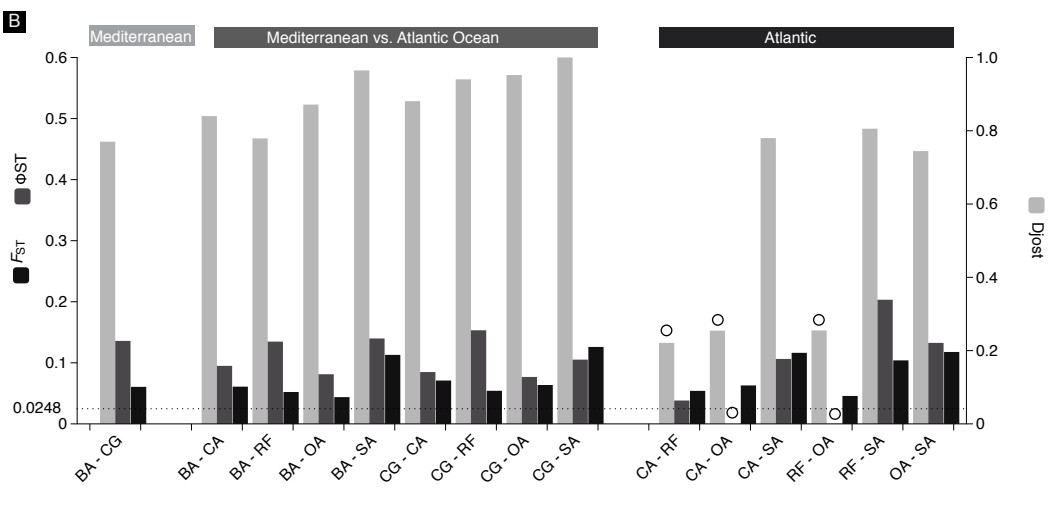

**Figure 5** $F_{ST}$, $\Phi_{ST}$ **and Djost pairwise values.** Mitochondrial differentiation $F_{ST}$, $\Phi_{ST}$ and $Dj$ost statistics. (A) Average values of the three geographical areas. (B) Between location pairs with $N > 10$. Location codes as in Fig. 2, BA, Barcelona; CG, Cabo de Gata; CA, Cadiz; OL, Olhos de Água; RF, Ria Formosa; SA, Sado. Significance of differentiation indicated with a circle was assessed through the calculation of 95% confidence limits using a bias corrected bootstrapping method. Line at 0.0248 indicates $\geq$95% confidence (95.5% Fisher's exact test, 96.2% chi-square) POWSIM threshold detection of $F_{ST}$.

distribution (ODD) and the expected distance distribution (EDD) of shared alleles does not reject the assumption of panmixia for the S7 dataset (ODD = 522 km, EDD = 509 km, $p > 0.62$).

# DISCUSSION

In this study, we evaluated four plausible phylogeographic scenarios to explain putative genetic differentiation between Mediterranean and Atlantic samples of *Salaria pavo* (Fig. 1). The nuclear marker with only 12 haplotypes, displayed comparatively low genetic diversity, probably due to low mutation rates (*Harpending, 1994*). From the haplotype network one can also clearly infer that there is no geographical structure. We will therefore discuss in more detail the mtDNA results. Pure models of panmixia, secondary contact, and presence of a phylogeographic break do not seem to explain the results obtained, while isolation-by-distance with asymmetric migration between the Atlantic and Mediterranean is a more plausible explanation. Before dissecting these results, it is appropriate to address two

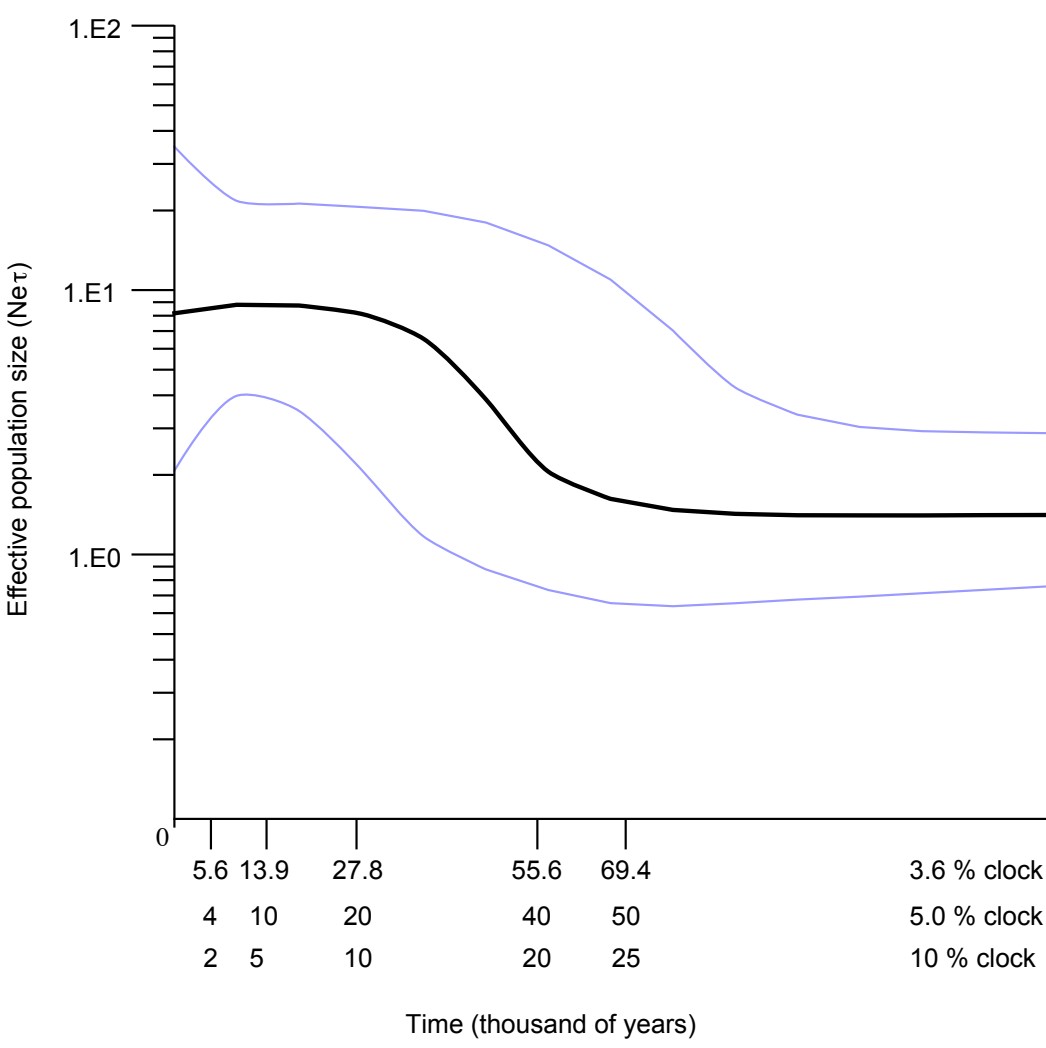

**Figure 6** **Bayesian skyline reconstructions showing the historical demographic trends for *Salaria pavo* for D-loop sequences.** Time, in thousands of years, is shown on the *x*-axis. Along the *y*-axis is the expressed population size estimated in units of $Ne\tau$ ($Ne$: effective population size, $\tau$: mutation rate per haplotype per generation). The central dark horizontal line in the plot is the median value for effective population size; the light lines are the upper and lower 95% HPD for those estimates.

main caveats regarding this work. Firstly, we are contrasting our results with well-defined hypotheses that constitute extremes of often less clear biological realities. Isolation-by-distance and clines are not mutually exclusive genetic patterns, as illustrated by ring species (*Irwin, Bensch & Price, 2001*), in which a series of intermediate subpopulations display a contact zone and are often connected by a cline at the closure of the ring (*Bensch et al., 2009*). Although the isolation-by-distance tests were only significant using the Jost's *D*, this is a sound metric to evaluate the genetic differentiation irrespective of haplotype diversity and genetic distance between populations (*Bird, Karl & Toonen, 2011*). Our results seem to reject panmixia, secondary contact and phylogeographic break models, and there is no evidence suggesting that a combination of these would be a better fit. Secondly, the nuclear

data display limited variability and no phylogeographic patterns could be identified, showing that S7 was not a good candidate gene for this particular species, although it has been successfully used in other marine fish (*Ahti et al., 2016*). Thirdly, the migration estimation must be taken with caution, because not all assumptions behind the method used were met (i.e., MIGRATE-N assumes a constant effective population size).

## Model evaluation

Panmixia (hypothesis 1) was concomitantly rejected by the haplotype network (Fig. 3), the spatial analysis of shared alleles (Fig. 4) and the MIGRATE-n results (Table 2). Moreover, the presence of private haplotypes detected in both Mediterranean and Atlantic locations and the fact that some of these were found multiple times on a single location suggests some limitations to gene flow (*Hartl & Clark, 1997*). Results regarding the classical isolation-by-distance regression model (hypothesis 2) were somewhat equivocal: rejection of the model based on $F_{ST}$ and $\phi_{ST}$, and non-rejection based on Jost's $D$. Because Jost's $D$ is independent from gene diversity, and it was shown to perform well in evaluating genetic differentiation regardless of haplotype diversity and genetic distance between populations (*Bird, Karl & Toonen, 2011*), we do not entirely reject the isolation-by-distance model. We found no support for hypothesis 3 (secondary contact) as most haplotypes are singletons or are shared between two locations, and no haplotype frequency cline could be detected. There is no evidence for a specific association between haplotype presence and locations, such as detected under a phylogeographic barrier (hypothesis 4), with all AMOVA among group $F_{CT}$ values returning non-significant. A modification of this hypothesis in which several groups are considered, was tested by implementing SAMOVA. The 4-group (BA)(CG)(CA-RF-OA)(SA) arrangement is the one that maximizes $F_{CT}$, when compared to all other possible groups (two, three and five-group arrangement), and it is also the one that that makes sense from the geographical and isolation-by-distance viewpoint. Barcelona, well in the Mediterranean constitutes a separate group; Cabo de Gata, on the frontier of the Alboran Sea, is also separate; Cadiz, Ria Formosa and Olhão, are all situated in the Atlantic region adjacent to the Strait of Gibraltar, forms another group; and finally, Sado a clearly Northeastern Atlantic location.

## The Atlantic-Mediterranean continuum and ancestral areas of refugia

Some fish species display a strong genetic discontinuity between each side of the Almeria-Oran oceanographic front, but this pattern is species-dependent. In the same family (e.g., Sparidae *Bargelloni et al., 2003*; *Galarza et al., 2009a*) and even within the same genus (e.g., *Diplodus*: *D. puntazzo* and *D. sargus*, *Bargelloni et al., 2005*), there are species with strong gene flow across the boundary, while others have restricted gene flow. *Salaria pavo* displays no significant differentiation across the Atlanto-Mediterranean boundary and this permeability contrasts with the strong across-boundary differentiation displayed by another intertidal blenniid *Coryphoblennius galerita* (*Francisco et al., 2014*). The strong thermohaline density gradient nature of the Almeria-Oran oceanographic front is apparently not sufficient to restrict the mobility of *S. pavo* across the boundary. On the other hand, the paleotemperatures estimated for the summer spawning season during

the last glacial maximum (LGM) were at most 13 °C in the Iberian Atlantic and most of the west Mediterranean (*LIMAP, 1981*) which are not compatible with the high thermal preferences of *S. pavo.* This species' embryos kept in laboratory arrest their development at temperatures of 15 °C or lower (*Westernhagen, 1983*). Considering these conditions, *S. pavo* was at the LGM most likely extirpated from its northern limit, the Bay of Biscay, as well as from North and Central Portugal. These locations represent postglacial colonizations derived from potential refugia located in the Mediterranean or further south in the Atlantic. The most northern location with a representative number of individuals (Sado) displays a significantly lower haplotype diversity values than those found in other Atlantic (Cadiz, Ria Formosa and Olhos d'Água) and Mediterranean locations (Barcelona), which is concordant with a postglacial colonization event.

The most probable refugium can be inferred by coalescent theory in which ancestral mitochondrial haplotypes are likely to have given rise to more derived ones because mutation has occurred over a longer period of time (*Posada & Crandall, 2001*). As a consequence, older haplotypes tend to have more connections in a network. Although homoplasy and high mutation rates can bias this pattern, highly connected haplotypes tend to be closely related to ancestral haplotypes (*Posada & Crandall, 2001*). Thus, the presence of highly connected haplotypes in the Atlantic could indicate this region as the likely major source of *S. pavo* post-glacial recolonization (Fig. 2). However, if gene flow persisted between Atlantic and west Mediterranean during the LGM, both areas may have operated as a vast refugium for the species. The hypothesis of a single refugium located inside the Mediterranean seems the least probable, but no definitive conclusions can be drawn.

The asymmetric gene flow detected in the peacock blenny is counterintuitive to expectations based on the prevalent out-of-Mediterranean surface currents (*Naranjo et al., 2015*). We posit that the unidirectional dispersal direction, also observed in other species (*Alberto et al., 2008*; *Xavier et al., 2011*) is disproportionally affected by sporadic storms that alter near-shore counter-currents (*Relvas & Barton, 2002*) and surface wind patterns rather than yearly or decadal averages of oceanographic conditions. Because the species lives and spawns in sheltered rocky habitats and coastal lagoons in the intertidal, or in the first meters of the subtidal, it is likely that costal water circulation patterns will affect more decisively its dispersion than the Atlantic-Mediterranean Sea general circulation model.

## Other factors contributing to the present pattern

*Salaria pavo* differs from other blennids by living preferentially in sheltered rocky habitats, estuaries and lagoons (*Zander, 1986*). Although the pelagic larval duration is of ca. 18 days at a temperature of 21 °C (*Westernhagen, 1983*), it seems likely that larvae from such sheltered habitats can be subject to more efficient retention than those of other blennids of more exposed shores. *Salaria pavo*'s differentiation pattern is consistent with a combination of considerable individual retention with sporadic episodes of range dispersal, which would reconcile the observation of high $D$, $F_{ST}$ and $\theta_{ST}$ values between locations separated by hundreds of kilometers with substantial sharing of haplotypes between other locations. However, one cannot discard the possibility of phylopatry or a recent limitation to gene-flow acting on a past panmitic population.

Previously published work hypothesized that the reduced genetic variation detected in *S. pavo* could have been the result of a severe bottleneck event (*Almada et al., 2009*). However, we have found a high number of widely distributed haplotypes, coupled with generally significant $F_{ST}$, $\phi_{ST}$ and *D*jost values for D-loop and an expansion signature, findings that do not support the hypothesis of a severe bottleneck. Results of the Bayesian skyline plot (BSP) of the *S. pavo* lineage A (Fig. 6) suggested a recent and rapid 100-fold increase in population size, preceded by a minor decrease that followed an extended period of stability. The lack of a species-specific clock and associated error requires cautious interpretation of age estimates, but the assumed rates of 3.6%, 5% and 10%/MY place the expansion unequivocally during the Pleistocene.

In summary, we propose that the genetic pattern of *S. pavo* in the Atlanto-Mediterranean region is better explained by a combination of some degree of isolation-by-distance and asymmetric migration. The ancestral lineage most probably originated in the Atlantic, where most of the genetic diversity is present. Both dispersal potential and physical factors such as local oceanographic conditions are playing a major role in shaping the genetic structure of this species.

## ACKNOWLEDGEMENTS

We are grateful to Sónia Chenu for her help with the lab work and to Marta Pascual and Ferran Palero, from the University of Barcelona, for their assistance in implementing the Salicrú test of haplotype diversity. Collection of specimens complied with the current laws of each country. We would like to acknowledge the contribution of the referees for this final version. We dedicate this paper to the loving memory of Prof. Vítor Almada who died during the course of this work.

### Funding

This study was funded by the MarinERA project "Marine phylogeographic structuring during climate change: the signature of leading and rear edge of range shifting populations"; by the Eco-Ethology Research Unit' Strategic Plan (PEst-OE/MAR/UI0331/2011), now included in MARE (UID/MAR/04292/2013), and by CCMAR Strategic Plan (PEst-C/MAR/LA0015/2011 and UID/Multi/04326/2013) from Fundação para a Ciência e a Tecnologia—FCT (partially FEDER funded). RLC was supported by a postdoctoral fellowship from FCT—Portuguese Science Foundation (SFRH/BPD/109685/2015). The funders had no role in study design, data collection and analysis, decision to publish, or preparation of the manuscript.

### Grant Disclosures

The following grant information was disclosed by the authors:
Eco-Ethology Research Unit' Strategic Plan: PEst-OE/MAR/UI0331/2011.
MARE: UID/MAR/04292/2013.

CCMAR Strategic Plan: PEst-C/MAR/LA0015/2011, UID/Multi/04326/2013.
FCT—Portuguese Science Foundation: SFRH/BPD/109685/2015.

## Competing Interests

Rita Castilho is an Academic Editor for PeerJ.

## Author Contributions

- Rita Castilho conceived and designed the experiments, performed the experiments, analyzed the data, contributed reagents/materials/analysis tools, wrote the paper, prepared figures and/or tables, reviewed drafts of the paper.
- Regina L. Cunha performed the experiments, analyzed the data, wrote the paper, reviewed drafts of the paper.
- Cláudia Faria performed the collection of samples.
- Eva M. Velasco reviewed drafts of the paper, collection of samples.
- Joana I. Robalo conceived and designed the experiments, performed the experiments, analyzed the data, contributed reagents/materials/analysis tools, wrote the paper, reviewed drafts of the paper, collection of samples.

## Animal Ethics

The following information was supplied relating to ethical approvals (i.e., approving body and any reference numbers):

No sampling permits were required as this species is listed as "least concern conservation status" and it was not captured in protected areas.

## Data Availability

GenBank:

HQ857214–HQ857383.

JF834709–JF834885.

## Supplemental Information

Supplemental information for this article can be found online at http://dx.doi.org/10.7717/peerj.3195#supplemental-information.

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
