# Peer review of "Asymmetrical dispersal and putative isolation-by-distance of an intertidal blenniid across the Atlantic–Mediterranean divide"

_PeerJ, doi:10.7717/peerj.3195_

## Round 0.1 · original submission · Major Revisions

Dear Dr. Castilho,

We have received the reports from three reviewers on your manuscript “Isolation-by-distance and asymmetrical dispersal of an intertidal blenniid across the Atlantic-Mediterranean divide”. See below the comments provided by the reviewers, which performed a detailed work. I must inform you that, based on the advice received, your manuscript may be considered acceptable for publication in PeerJ after a major review is performed. Please, give particular attention to concerns regarding the sampling design and validity of findings (see alternative hypothesis for the isolation-by-distance with asymmetrical gene flow hypotheses given by reviewer 1). Please, consider all the referees’ remarks in your revised manuscript and indicate in your rebuttal letter a point-by-point response to the referees
.
With kind regards,

Ronaldo

Reviewer 1 ·

Basic reporting

The manuscript “Isolation-by-distance and asymmetrical dispersal of an intertidal blenniid across the Atlantic- Mediterranean divide” aims to understand the phylogeographic and evolutionary history of Salaria pavo. The manuscript is well-written, the sentences are coherent and the premise of interest.
There are some minor corrections that need to be addressed throughout the text, and species names should always be italicised.
I have also suggested extra references, regarding the use of mutation rates and BSP inferences that I think would greatly benefit the manuscript.
(see comments below)

Experimental design

The sampling scheme of the manuscript is accurate for the proposed study. However, the Migrate-N and BSP analyses need to be addressed.
I would strongly recommend for the authors to use IMA2 instead of Migrate-N, as well as other faster mutation rates (see comments below).

Validity of the findings

The authors conclude that the most likely model of genetic differentiation in Salaria pavo is an isolation-by-distance with asymmetrical gene flow, from the Mediterranean to the Atlantic Ocean. I’m not certain I agree with these findings, as it seems that panmixia is a strong possibility.
Some of the analyses need to be re-assessed and better explained to substantiate this claim.
(see comments below).

Additional comments

The manuscript “Isolation-by-distance and asymmetrical dispersal of an intertidal blenniid across the Atlantic- Mediterranean divide” aims to understand the phylogeographic and evolutionary history of Salaria pavo. Using a combination of mtDNA and nDNA, the authors investigated historical levels of genetic differentiation and gene flow, and tested the validity of four different evolutionary scenarios. I quite enjoyed the way this was presented in the introduction and methods, but I have some reservations regarding the results and their interpretation. Therefore, I recommend the manuscript to be accepted pending major revisions.
Major comments:
- The recent nature of the population expansion in S. pavo suggests that the species has yet to reach mutation-drift equilibrium. Therefore, Migrate-N might not be the best software to investigate levels of gene flow among sites. I would strongly recommend the use of IMA2. Furthermore, it is my impression that FST levels were quite low (although these were not easily available in the text), which might also impair the estimates of gene flow with Migrate-N
- Figures 4 and 5 are quite interesting, but difficult to follow. These represent the crucial findings of the manuscript, but it took me awhile to understand them. I would recommend adding a table with genetic divergence levels (even if in annex)
- Is there biased dispersal in this species? It might be one of the reasons behind the difference between mtDNA and S7. Although the slower mutation rate of the latter could also account for the observed patterns, the possibility of philopatry should be mentioned
- The Bayes Factor analyses (Table 2), seem to suggest that Model 1 (panmixia) had the highest values and was statistically significant. Unless I’m reading the table wrong, this would point to panmixia as the most likely model (which would agree with the findings of Figure 5). Please elaborate.
- I recommend re-running BSP analyses with a higher mutation rate (e.g. u=10%, see Bowen et al. 2006 and Grant 2015) – this is likely to change the results and final interpretations.

Minor comments:
Line 47: Consider re-wording this sentence
Line 52: use impacting instead
Line 54: the contiguous
Line 60: species names should be italicised
Line 67: use mainly instead
Line 73 and throughout the text: replace breeding with spawning
Line 79: replace in with from
Line 103: remove preserved in 96% ethanol
Line 107: not according to Table 1, where the numbers between mtDNA and S7 do not always match. Add: whenever possible
Line 130: In Table 1 there is also heterozygosity – either remove it or add it here
Line 136: POWSIM analyses need to be better described: this was not the only FST level tested (according to your results), and all parameters should be included – Ne, number of generations and sample size
Line 137: explain the rationale of using Jost’s D? It is generally recommended for multi-character states loci, such as microsatellite markers.
Line 161: Salaria
Line 183 and 186: add the versions of these softwares (since all others had versions)
Line 192: I would suggest to also re-run these analyses with u=10% per Million years, see:
Bowen BW, Muss A, Rocha LA, Grant WS (2006) Shallow mtDNA coalescence in Atlantic Pigmy angelfishes (Genus Centropyge) indicates a recent invasion from the Indian Ocean. Journal of Heredity, 97: 1-12.
Grant WS (2015) Problems and cautions with sequence mismatch analysis and Bayesian Skyline Plots to infer historical demography. Journal of Heredity, 106, 333-346.
Line 208: if they were not significantly different, then p>0.05
Line 211: remove the explanation about a star-shaped network.
Line 219-221: This is not what it is mentioned in the methods: Line 135 - FST = 0.05. If other FST values were tested (which I agree with), then they should be mentioned in the methods section.
Line 223-231: Figures 4 and 5 are interesting, but I would like to see the overall FST/D values, and the one for the Mediterranean - Atlantic comparison
Line 255: Do the authors mean p>0.62?
Line 258: scenarios to explain putative genetic differentiation
Lines 263-266: Table 2 appears to suggest panmixia as the most likely model. Please explain.
Line 273-274: other explanations include philopatry and that the possible break in gene flow is rather recent, and in the ancient past the species were composed by one panmitic population. These should be included in the discussion
Line 280: see comment above regarding table 2
Line 295: species names should be italicised
Line 303: spawning
Line 303: in the Iberian Atlantic
Line 304: which are not compatible
Line 308-309: Which would suggest that the species is not in mutation-drift equilibrium and, as such, Migrate-N is not the most adequate software. I would recommend the authors to re-do that section using IMA2
Line 310: In the results, the authors said that there were no significant differences among haplotype diversity levels from different locations. Therefore, this sentence needs to be re-written.
Line 320: replace the Atlantic by this region
Line 324-325: why? Please elaborate
Line 334: I cannot infer this from the results presented. I would like to see a FST table
Line 339-340: why?
Line 348: FST in subscript
Line 377: incomplete reference

Reference list: remove all dois, or include them in all references

Figure 1: I quite like this figure! It is very useful (particularly for an educational perspective - think I will use it in future lectures)

Figure 2: There are no numbers in parenthesis in the figure; Consider rephrasing this legend, as it does not accurately match the figure. I cannot understand what are the large and small pies with dark blue and light blue.
Also, add a key for the haplotype colour
Furthermore, I could not access Annex 1

Figure 3: remove network from the legend and add a scale to show the proportionality of mutations

Table 1: it would be better to show nucleotide diversity the classical way: 0.06 - which is actually quite high, and very different from the other sites. Please, check if it is correct.

Table 2: If I'm reading this table correctly, the panmixia model has the higher Log Bayes Factor and P < 0.05. This would suggest that panmixia is the most likely scenario for this species, with a possible break in gene flow and asymetrical migration

Reviewer 2 ·

Basic reporting

The article is well written and well organized. Included figures and tables are appropriate and clearly presented, with a few exceptions I've noted in my general comments to the authors.

Experimental design

Sampling at several sites is insufficient (Formentera and Galacia) and these sites should therefore not be included in tests for population structure. As I note below, I recommend testing for population structure in the S7 locus or adding additional loci to corroborate dloop findings. I as well recommend hierachical analysis of population structure for Dloop (and S7) to test for regional and subpopulation structure (which is indicated in figures 2 and 3).

Validity of the findings

As a single locus study, I have a hard time definitively accepting the authors' conclusion of assymetrical IBD. Dloop is notoriously challenging to work with because of it's hypervariability, which bias Fst estimates and considerably reduce power. See my general comments to the authors for a more detailed description of my concerns and recommendations.

Additional comments

The article submitted by Castilho and colleagues investigates connectivity and demographic history of peacock blenny among sites spanning the boundary between the Mediterranean and Atlantic segments of the Lusitania biogeographic province. The phylogeographic significance of this boundary is currently unclear, with some species exhibiting a genetic differentiation at the Strait of Gibraltar, while others appear well mixed (reviewed in Paternello et al. 2007). Because of it’s low vagility and relatively short larval phase, genetic assessment of the peacock blenny provides an opportunity for a robust test for historic isolation across the Strait of Gibraltar. The paper is well written and the questions addressed have the potential to offer valuable insights into fine-scale biogeographic and evolutionary processes. However, because the authors elected to not analyze the s7 locus, they effectively present a single locus study with associated ambiguity in findings. I therefore recommend analysis of s7 data since even small shifts in allele frequencies among sites or regions can help resolve higher-level population structure. Alternatively, if the authors feel S7 is truly uninformative then I recommend dropping it from the study and including one or more additional loci to corroborate findings from dloop. With regard to dloop, I agree with the bulk of the analyses but I think an additional hierarchical analysis of structure (AMOVA) may be informative. While the authors may be correct in identifying IBD as the dominant pattern, figures 2 and 3 suggest a more complicated pattern, with sites clustering by regions due to historic and/or current barriers to gene flow. Interestingly, the authors seem to recognize this regional structuring of the populations in figure 2 but no mention of it is made in the text or figure 2 caption. In addition, sample sizes are low for many of the sites, particularly given the hypervariability of the dloop locus. I would at a minimum recommend dropping Formentera (n=2) and Galicia (n=1) from all analyses if sample sizes cannot be increased.

Specific comments
• Line 54: “Atlantic Ocean staged” is unclear
• Line 60: “strong genetic flow” seems a strange turn of phrase, perhaps “high gene flow” would more clearly make the intended point
• Line 78: This is the first mention of “LGM”. Please define the abbreviation and as well indicate why S. pavo were likely absent from the region during the LGM. Clearly you’re indicating regional temperatures were below S. pavo’s current tolerance, so you just need a sentence or two indicating expected LGM temperatures.
• Line 86-94: I like this setup with a list of the various possible scenarios; however, while I realize much of the project is centered on testing for structure across the boundary between the Med. and Atlantic, given the inconsistency in response to the “barrier” seen in other species, and the complex population structure often observed in blennies, I think an alternative hypothesis of >2 populations needs to be included. Maybe you could account for this possibility with a slight modification of hypothesis 4, indicating something to the effect of a sharp genetic break between regions with additional subpopulation structure within regions?
• Line 137: Kudos for including Jost’s D!
• Population structure in general: As I mention above, I suggest testing for population structure in S7 as well and, for both loci, adding AMOVA, which can be run on Fst and D in GenAIeX (Peakall and Smouse 2012). AMOVA offers an opportunity to test for hierarchical structure. An overall AMOVA (with no groupings) would indicate whether there’s panmixia. The next obvious grouping would be Med. versus Atlantic, which I’m guessing will return significant Fct and/or Dct (among groups) and Fsc and/or Dsc (within groups) values. However, other groupings might return higher among group and lower within group estimates of differentiation. I would therefor as well suggest testing several possible scenarios using significant pairwise results to determine likely groupings, or by running a PCA analysis (Patterson et al. 2006) using haplotype frequencies at each site and then running AMOVA on any apparent clusters of sampling sites.
• Line 145: I’m unfamiliar with this analysis. Does it have more power to detect deviations from panmixia and Fst or D?
• Lines 146-147: shared haplotypes may reflect historical processes and not current gene flow. You could acknowledge this by simply adding “evidence of recent or ongoing gene flow.”
• Line 189-192: It is unclear whether 3.6% or 5% mutation rate was used
• Lines 223-229: Please somehow indicate significance of Fst and Dst comparisons. Perhaps simply adding a “*” to significant comparisons in figure 5 would be sufficient?
• Line 230: Was PHIst run instead of Fst? Given your observed dloop diversity, there’s some justification for running PHIst rather than just considering haplotype frequencies (Fst). That said, I would more trust Dst because of the inherent negative bias in fixation based estimates of population structure with increasing intraspecific diversity.
• Lines 234-239: Please provide estimated number of migrants and theta.
• Line 238: typo “ca.”
• Line 245: typo “using 5% dates at 200,00”
• Lines 252-253: typo “does evidence”
• Lines 326-327: You've indicated bayes factors support higher out of Mediterranean gene flow, which would appear to be consistent with prevailing surface currents.
• Line 336: Because Formentera n=2, any related Fst and Dst estimates are unreliable
• Line 342 and 348: PHIst or Fst?
• Figure 2: What are the purple lines?
• Figure 4: If including the figure key please clearly indicate what each “expected” and “observed” symbol represents
• Figure 5: Typo “Gst”?
• Figure 5: What does figure 5a depict?
• Figure 6: Grey rectangle corresponding to LGM not shown in figure

References:

Paternello, T., Volckaert, A.M.J. & Castilho, R. (2007) Pillars of Hercules: is the Atlantic–Mediterranean transition a phylogeographic break? Molecular Ecology, 16, 4426–4444.
Patterson N, Price AL, Reich D. Population Structure and Eigenanalysis. Allison DB, ed. PLoS Genetics. 2006;2(12):e190. doi:10.1371/journal.pgen.0020190.

Peakall, R. and Smouse P.E. (2012) GenAlEx 6.5: genetic analysis in Excel. Population genetic software for teaching and research-an update. Bioinformatics 28, 2537-2539.

·

Basic reporting

The manuscript is pretty clearly laid out. It has a good structure, providing a good background, and also has very nice and clear figures.
At the end of the Introduction the criterion for choosing the species should be more openly stated. The Discussion, on the other hand, over-emphasizes IbD (see further below).

Experimental design

Fair design. samples allow for hypothesis testing.
information on fish euthanisation or live-handling should be clarified.

Validity of the findings

The findings are generally valid and realistic, with the caveat that the authors over-emphasize the role of Isolation by Distance, while their main test to detect it (Mantel) actually fails to reject the null hypothesis.

Additional comments

I am attaching the pdf with a few comments throughout the text.
I recommedn the authors to do as follows:
1) try a Mantel test simply based on an Fst matrix, as opposed to a Phi-st, as that way, historical polymorphisms will not affect the wide intra-sample nucleotide variance. This way, using only frequency-based differentiation, it may help picking up some association between genetic and geographic matrices.
2) unless point 1 revolutionises the relationship, I would considerably tone down the role of IbD, because it is somewhat speculative/hazy.

---

## Round 0.2 · Minor Revisions

As you can see, two of the original reviewers have re-reviewed your submission. Although the comments of Reviewer 1 are mostly 'editorial', Reviewer 2 still has a significant concern which (as they indicate) could be addressed via some additional analyses.

Please address all of the review comments in your revision.

Reviewer 1 ·

Basic reporting

The revised version of the manuscript "Isolation-by-distance and asymmetrical dispersal of an intertidal blenniid across the Atlantic-Mediterranean divide" is much improved, with better background and context provided, and a more balanced discussion.
I would suggest changing the title to reflect this version, though.

Experimental design

Experimental design is adequate and the new analyses done provide further support to the findings.

Validity of the findings

The new analyses done and the changes introduced in the manuscript make it more robust.

Additional comments

The authors of "Isolation-by-distance and asymmetrical dispersal of an intertidal blenniid across the Atlantic-Mediterranean divide" are to be congratulated for their effort in answering all my comments and addressing my concerns. The new version of the manuscript is much improved and provides a more balanced interpretation of the results. I am particularly pleased with the thorough reply considering the IMA-2 and Migrate-N discussion, and I have no further comments on that part. Therefore, I recommend the manuscript to be accepted pending minor (mainly editorial) revisions.

Minor changes:
Title - the new version of the manuscript acknowledged the difficulty in attributing the observed phylogeographic pattern in Salaria pavo to a specific model. Therefore, I would recommend for the title to be changed to accommodate the new version.
Line 67: add a - between sargus and Bargelloni
Line 89: Spawn (and not breed)
Line 121: were observed - remove be
Line 122: returned to the same pool
Line 127: remove ( from 1998
Line 132: not necessary to show the primer sequences if they are described elsewhere
Line 168: and phiST
Line 168: remove and, make it two sentences
Line 172: add a ,
Line 198: missing a ) at the end
Line 232: add a .
Line 226: it seems that BSP analysis was only conducted for the Atlantic samples, please specify this in the methods. Otherwise, the authors should have estimated BSP for both populations
Line 243: replace characters by bp
Line 250: replace eleven by 11
Line 253: a nucleotide diversity of 0.036 is not low. Please change accordingly
Line 257: remove )
Line 277 (and 364): Isolation-by-distance
Line 286: remove the extra sentence saying AMOVA results
Line 312: a nucleotide diversity of 0.18 is not low. Was this in %?
Line 381: add (LGM) after last glaciation maximum

Figure 5: Not sure why it says GST, when it should be FST and phiST. Also, what are the circles in the figure? I'm assuming they mean non-significant, but that does not show in the legend, which mentions asterisks.

Reviewer 2 ·

Basic reporting

The revised manuscript is well written and well structured, and figures/tables are clear and appropriate.

Experimental design

I appreciate the significant time and effort the authors put into revising the manuscript, including undertaking additional analyses of population structure and a more detailed assessment of the S7 data. That said, I do not see any indication of results for a pairwise analysis of population structure nor the related IBD analysis for the S7 data. Likewise, AMOVA was not conducted using Dest for either loci.

I feel this is a non-trivial point as the argument for IBD is based on a significant signal in the D-loop Dest data, yet IBD was not tested using the S7 data and AMOVA/SAMOVA were not run using Dest.

I recommend the authors look into Genodive (Meirmans and Van Tienderen 2004) which offers an ability to test for hierachical structure (AMOVA) using several standardized measures of differentiation (including Dest).

Meirmans, P.G., and P.H. Van Tienderen: (2004), GENOTYPE and GENODIVE: two
programs for the analysis of genetic diversity of asexual organisms, Molecular Ecology
Notes 4 p.792-794.

Validity of the findings

While I agree with many of the authors' conclusions, my concern is that without additional analyses I feel that there is only marginal support for the IBD model.

Additional comments

While I think the manuscript will be well received once published, I feel there is still some ambiguity regarding the importance of IBD versus regional population structure. As I've indicated above, I feel the manuscript would benefit from a few additional analyses. Since the conclusion of IBD was based in part on a significant IBD pattern in the d-loop Dest data set, I think it would be appropriate to test this same data set for hierarchical population structure (AMOVA) using Genodive. I as well think it would be informative to test pairwise population structure and IBD (mantel test) using the S7 dataset, both of which can be run in Arlequin.

---

## Round 0.3 · accepted · Accept

This article is Accepted. In the previous review round, Reviewer 1 indicated they would be satisfied with the minor edits and would not need to see the manuscript again. This version has been re-reviewed by Reviewer 2 who now also finds it Acceptable.

Reviewer 2 ·

Basic reporting

No further comment

Experimental design

No further comment

Validity of the findings

I very much appreciate the author's effort to address my concerns regarding what I felt were some inconsistencies regarding Dloop and S7 analyses, and I apologize for recommending analyses that were at this point not possible. I was under the impression that AMOVA could be conducted using Jost Dest in Genodive; however, I was clearly mistaken.

Additional comments

I feel the revised manuscript appropriately acknowledges the tentative support for the IBD model, and I am pleased to recommend publication of the manuscript in its current form.

---

## Author Rebuttal · Round 0.3

Rita Castilho
CCMAR
University of Algarve
Campus de Gambelas
8005-139 Faro
Portugal

Dear Editor,

We are very thankful for the opportunity offered to resubmit to *PeerJ* our manuscript entitled " **Asymmetrical dispersal and putative isolation-by-distance of an intertidal blenniid across the Atlantic-Mediterranean divide**" by Castilho et al. that we have now uploaded a revised version to comply with the minor revisions decision.

All suggested changes were taken into consideration, and to the best of our ability either incorporated in the manuscript or justified their non-inclusion. Additionally, we have amended the abstract for better reading. Please find below interspersed with the reviewer's comments our detailed reply, which we have highlighted using a blue font color. We would like to thank all referees, but especially referee #1, for his thorough and relevant contribution.

We hope to have answered all raised issues to your satisfaction. We look forward to hearing from you.

Yours sincerely,

Rita Castilho

Editor's Comments
MAJOR REVISIONS

As you can see, two of the original reviewers have re-reviewed your submission. Although the comments of Reviewer 1 are mostly 'editorial', Reviewer 2 still has a significant concern which (as they indicate) could be addressed via some additional analyses.

Please address all of the review comments in your revision.

## Reviewer 1 (Anonymous)

### Basic reporting

The revised version of the manuscript "Isolation-by-distance and asymmetrical dispersal of an intertidal blenniid across the Atlantic-Mediterranean divide" is much improved, with better background and context provided, and a more balanced discussion.

**Reply:** We thank the referee for his positive comment.

I would suggest changing the title to reflect this version, though.

**Reply:** The title was change to incorporate the equivocal evidence from IBD.

### Experimental design

Experimental design is adequate and the new analyses done provide further support to the findings.

**Reply:** We thank the referee for this positive comment.

### Validity of the findings

The new analyses done and the changes introduced in the manuscript make it more robust.

**Reply:** We thank the referee for this positive comment.

### Comments for the Author

The authors of "Isolation-by-distance and asymmetrical dispersal of an intertidal blenniid across the Atlantic-Mediterranean divide" are to be congratulated for their effort in answering all my comments and addressing my concerns. The new version of the manuscript is much improved and provides a more balanced interpretation of the results. I am particularly pleased with the thorough reply considering the IMA-2 and Migrate-N discussion, and I have no further comments on that part. Therefore, I recommend the manuscript to be accepted pending minor (mainly editorial) revisions.

Minor changes:
Title - the new version of the manuscript acknowledged the difficulty in attributing the observed phylogeographic pattern in Salaria pavo to a specific model. Therefore, I would recommend for the title to be changed to accommodate the new version.

**Reply:** We thank the referee for the suggestion and we introduced a nuance in the present title: "Asymmetrical dispersal and putative isolation-by-distance of an intertidal blenniid across the Atlantic-Mediterranean divide"

✓ Line 67: add a - between sargus and Bargelloni
✓ Line 89: Spawn (and not breed)
✓ Line 121: were observed - remove be
✓ Line 122: returned to the same pool
✓ Line 127: remove ( from 1998

✔ Line 132: not necessary to show the primer sequences if they are described elsewhere

✔ Line 168: and phiST

✔ Line 168: remove and, make it two sentences

✔ Line 172: add a ,

✔ Line 198: missing a ) at the end

✔ Line 232: add a .

✔ Line 226: it seems that BSP analysis was only conducted for the Atlantic samples, please specify this in the methods. Otherwise, the authors should have estimated BSP for both populations

**Reply:** We have made that explicit in the Material and Methods section.

✔ Line 243: replace characters by bp

✔ Line 250: replace eleven by 11

✔ Line 253: a nucleotide diversity of 0.036 is not low. Please change accordingly

✔ Line 257: remove )

✔ Line 277 (and 364): Isolation-by-distance

✔ Line 286: remove the extra sentence saying AMOVA results

✔ Line 312: a nucleotide diversity of 0.18 is not low. Was this in %?

**Reply:** It is percentage, the sign % was added.

✔ Line 381: add (LGM) after last glaciation maximum

✔ Figure 5: Not sure why it says GST, when it should be FST and phiST. Also, what are the circles in the figure? I'm assuming they mean non-significant, but that does not show in the legend, which mentions asterisks.

**Reply:** Gst was changed to Fst, and asterisks were change by circles in legend.
* * *
Reviewer 2 (Anonymous)

**Basic reporting**

The revised manuscript is well written and well structured, and figures/tables are clear and appropriate.

**Reply:** We thank the referee for this positive comment.

**Experimental design**

I appreciate the significant time and effort the authors put into revising the manuscript, including undertaking additional analyses of population structure and a more detailed assessment of the S7 data. That said, I do not see any indication of results for a pairwise analysis of population structure nor the related IBD analysis for the S7 data.

**Reply:** Results were included in file "Salaria_Appendix S1.xlsx", in "Pairwise values" sheet. A sentence was now also added to the results section. "Also pairwise comparisons between locations, which returns three significant values, always involving Ria Formosa location (Ria Formosa – Cabo de Gata; Ria Formosa – Cadiz; Ria Formosa - Sado) (Appendix S1) and non-significant values for the isolation-by-distance Mantel test (Appendix S1)."

Likewise, AMOVA was not conducted using Dest for either loci.
I feel this is a non-trivial point as the argument for IBD is based on a significant signal in the D-loop Dest data, yet IBD was not tested using the S7 data and AMOVA/SAMOVA were not run using Dest.
I recommend the authors look into Genodive (Meirmans and Van Tienderen 2004) which

offers an ability to test for hierachical structure (AMOVA) using several standardized measures of differentiation (including Dest).

Meirmans, P.G., and P.H. Van Tienderen: (2004), GENOTYPE and GENODIVE: two programs for the analysis of genetic diversity of asexual organisms, Molecular Ecology Notes 4 p.792-794.

**Reply:** We appreciate this suggestion and we would like to implement it; however, we have contacted the author, Patrick Meirmans, and he confirmed that Genodive does not have that capability. (see email below).

[Figure]

From: **Patrick Meirmans** p.g.meirmans@uva.nl
Subject: Re: AMOVA with D
Date: 28 February, 2017 at 13:38
To: Rita Castilho rcastil@ualg.pt

Dear Rita,

You have to disappoint your reviewer, because it is not possible at all to calculate Djost using an AMOVA. I have been thinking of ways to do this, and actually could not come up with a solution, so I think that it may not be possible at all.

What you can do is to use GenoDive to calculate D based on heterozygosities. However, that only works for a single population level, not when the populations themselves are clustered into groups (as one often does in an AMOVA).

Cheers, Patrick

> On 28 Feb 2017, at 08:58, Rita Castilho <rcastil@ualg.pt> wrote:
>
> Dear Patrick
> A reviewer insists I do an AMOVA with Djost, and suggested your software. However, I do not find a straightforward way of performing this analysis, can you please give me some directions?
>
> Thanks,
> Rita

**Validity of the findings**
While I agree with many of the authors' conclusions, my concern is that without additional analyses I feel that there is only marginal support for the IBD model.
**Reply:** We agree with the referee, and the text was modified in the previous submission to tone down the IBD claim. Also, the new title, as suggested by referee 1 also reflects a cautionary approach to the IBD claim.

**Comments for the Author**
While I think the manuscript will be well received once published, I feel there is still some ambiguity regarding the importance of IBD versus regional population structure. As I've indicated above, I feel the manuscript would benefit from a few additional analyses. Since the conclusion of IBD was based in part on a significant IBD pattern in the d-loop Dest data set, I think it would be appropriate to test this same data set for hierarchical population structure (AMOVA) using Genodive. I as well think it would be informative to test pairwise population structure and IBD (mantel test) using the S7 dataset, both of which can be run in Arlequin.
**Reply:** We have done an exhaustive search for alternative software that would implement the approach suggested by the referees, but we could not find any such software. As suggested, we have performed the pairwise population structure and IBD test (as referred above).